# Anodic TiO_2_ Nanotube Layers for Wastewater and Air Treatments: Assessment of Performance Using Sulfamethoxazole Degradation and N_2_O Reduction

**DOI:** 10.3390/molecules27248959

**Published:** 2022-12-16

**Authors:** Marcel Sihor, Sridhar Gowrisankaran, Alexandr Martaus, Martin Motola, Gilles Mailhot, Marcello Brigante, Olivier Monfort

**Affiliations:** 1Department of Inorganic Chemistry, Faculty of Natural Sciences, Comenius University Bratislava, Ilkovicova 6, Mlynska Dolina, 84215 Bratislava, Slovakia; 2Institute of Environmental Technology, CEET, VSB-Technical University of Ostrava, 17. Listopadu 15/2172, 70800 Ostrava-Poruba, Czech Republic; 3Institut de Chimie de Clermont-Ferrand, Université Clermont Auvergne, CNRS, Clermont Auvergne INP, F-63000 Clermont-Ferrand, France

**Keywords:** photocatalysis, pharmaceutical, water treatment, air treatment, N_2_O, TiO_2_

## Abstract

The preparation of anodic TiO_2_ nanotube layers has been performed using electrochemical anodization of Ti foil for 4 h at different voltages (from 0 V to 80 V). In addition, a TiO_2_ thin layer has been also prepared using the sol–gel method. All the photocatalysts have been characterized by XRD, SEM, and DRS to investigate the crystalline phase composition, the surface morphology, and the optical properties, respectively. The performance of the photocatalyst has been assessed in versatile photocatalytic reactions including the reduction of N_2_O gas and the oxidation of aqueous sulfamethoxazole. Due to their high specific surface area and excellent charge carriers transport, anodic TiO_2_ nanotube layers have exhibited the highest N_2_O conversion rate (up to 10% after 22 h) and the highest degradation extent of sulfamethoxazole (about 65% after 4 h) under UVA light. The degradation mechanism of sulfamethoxazole has been investigated by analyzing its transformation products by LC-MS and the predominant role of hydroxyl radicals has been confirmed. Finally, the efficiency of the anodic TiO_2_ nanotube layer has been tested in real wastewater reaching up to 45% of sulfamethoxazole degradation after 4 h.

## 1. Introduction

With regard to a number of publications, titania (TiO_2_) is the most investigated photocatalyst in a multitude of applications such as, for example, antibacterial coatings and water and air treatments [1,2,3,4,5,6]. Efficient TiO_2_ photocatalysts are prepared in the form of nanomaterials ranging from 0D to 3D morphologies [5,7,8,9,10,11,12]. The 1D titania nanostructures include TiO_2_ nanotubes (TNT) that are promising nanostructured photocatalysts mainly due to their excellent electron transport [13,14,15]. In the form of self-organized arrays (i.e., supported layers), TNT exhibit enhanced charge carriers separation, thus leading to exceptional photocatalytic properties [13,14,15,16]. The preparation TNT layers can be conducted by anodic oxidation of a Ti electrode in a fluoride-containing electrolyte [17,18,19]. The variation in applied voltage, fluoride concentration and anodization time can be tuned to design TNT of specific length and wall thickness [17,18,19,20]. In addition, by extending one of these experimental parameters, the TNT morphology can be transformed in a porous nanostructure, which also exhibits interesting photocatalytic properties as reported in our recent study [20]. Indeed, by increasing the anodization voltage, it is possible to form a porous sponge-like structure.

The treatment of water by photocatalysis is one of the most investigated alternatives to enhance the already existing processes which are used in wastewater treatment plants (WWTPs) and in the production of drinking water [21]. Indeed, the photocatalytic process belongs to advanced oxidation processes (AOPs), thus generating reactive oxygen species (ROS) such as hydroxyl radicals (HO•) [22]. Using a semiconductor photocatalyst such as TiO_2_, UVA light is necessary to generate the electron/hole (*e*^−^/*h*^+^) pair, from which *h*^+^ reacts with water to produce HO•. The HO• react with high kinetic rate and non-selectively with organic molecules including persistent organic pollutants (POPs) and contaminants of emerging concern (CECs) [22]. Among CECs, pharmaceutical and personal care products (PPCPs) are consumed daily and their long-term impact on the natural environmental and human health is not yet clarified [23,24]. Intergovernmental agencies have implemented stringent norms for pollution control and water quality, such as the implementing decision of the European Union (EU) No. 2020/1161 on the directive No. 2008/105/CE [25]. Therefore, the modification of conventional water treatments appears a necessary conditions to fulfill these norms, especially by adding or replacing current chemical oxidation processes with AOPs; thus, TNT photocatalysts are an excellent candidate. In this study, sulfamethoxazole (SMX) is used as a model pollutant since it is a widely used antibiotic. In addition, SMX is one of the most frequently detected pollutants in water around the world [23].

The treatment of air for the removal of nitrous oxide (N_2_O) using photocatalytic reactions was described back in the 1990s, when the first experiments employed zeolite containing Cu-based photocatalysts [26]. N_2_O is in the top 3 gases responsible for global warming, since it has been proven to be involved at 6.2% of the total global radiative forcing. The major origin of N_2_O is from natural processes (nitrification of ammonia, denitrification of nitrates, etc.) but also anthropogenic activities (N-based fertilizers, combustion of fossil fuels, etc.) [27]. Therefore, intense research is being carried out to develop different types of photocatalysts including TiO_2_ [28,29,30,31,32,33]. However, one significant disadvantage of these already existing photocatalysts is that they are not easy to handle because of their powder form. Using supported TNT photocatalysts, such a drawback might be overcome.

In the present study, the use of anodic TNT layers in wastewater and air treatments is assessed using the degradation of aqueous sulfamethoxazole solution and the reduction of gaseous nitrous oxide, respectively, as model reactions. To the best of our knowledge, it is the first time that anodic TiO_2_ layers either in the form of nanotubal or sponge-like structure are used in such versatile photocatalytic reactions, and their performance is compared to another TiO_2_ nanostructure i.e., TiO_2_ thin layer deposited by the sol–gel method. Such a comparative study will bring further insights into the application of TiO_2_ nanomaterials for environmental purposes. In addition, the degradation of SMX is also performed in real wastewater and the degradation pathway of SMX is proposed, so the fundamental understanding of the prepared materials is coupled with semi-applied science, which is the early stage of industrial application development. 

## 2. Results and Discussion

### 2.1. Structural and Optical Properties of Photocatalysts

The XRD patterns (Figure 1) of the F20, F40, F60, F80 and SG samples match with the anatase phase of TiO_2_ (ICDD card no. 03-065-5714) as reported in our previous studies [20,34]. Very weak diffractions of TiO_2_ anatase phase are also confirmed in the SG sample, where a textured silicon wafer was used as the substrate. Therefore, additional diffractions of the textured silicon wafers correspond to the *K_beta_*, *L_alpha1_* and *L_alpha2_* lines of Si (111) along with impurities. It is worth noting the diffractions from metallic Ti in the TNT layers since it is the underlying substrate of self-organized and highly oriented TiO_2_ nanotubes. In addition, the F0 sample only exhibits metallic Ti diffraction (ICDD card no. 00-044-1294), although an ultra-thin layer of TiO_2_ (several nanometers) might be present due to surface oxidation. 

The surface and cross-section morphology of the samples have been characterized by SEM in previous studies [20,34,35] and comprehensively discussed. Briefly, the surface of the F0 sample exhibits the typical morphology of Ti, i.e., a dense structure of microcrystals. As the anodization voltage increases from 20 to 80 V, a porous nanostructure is formed. The F20 sample exhibits the clear nanotubular morphology of a self-organized and highly-ordered TNT layer, while the surface of the TNT in the F40 sample starts to be destroyed. In the F60 and F80 samples, the nanotubular morphology disappears and a porous nanostructure composed of a reminiscent nanotube and a sponge-like structure is formed, respectively.

Concerning the optical properties, the Tauc’s plot (Figure 2) exhibits that the samples showing TNT morphology i.e., F20 and F40, possess the strongest light absorption, along with *E_g_*, of about 3.28 eV. For porous nanostructures such as the F60 and F80 samples, light absorption decreases while the *E_g_* is about 3.35 eV. For the SG sample, light absorption is weaker due to the thickness, which is approx. 5 times thinner than anodized samples. In addition, the F0 sample which is essentially composed of metallic Ti exhibits slight absorption in UVA due to the upper oxidized layer. 

### 2.2. Reduction of N_2_O

The ability of the TiO_2_ photocatalysts prepared by two different methods (electrochemical anodization and sol–gel deposition) to reduce nitrous oxide under UVA light is investigated (Figure 3). It is worth noting there is no reference in the literature about testing the photocatalytic decomposition of N_2_O with the use of a TiO_2_ nanostructure prepared by electrochemical anodization, thus highlighting the significance of the present study. 

From Figure 3, it is clear that N_2_O conversion, which occurs during photocatalytic reactions, is significantly higher than direct photolysis (i.e., without photocatalyst). Comparing the different photocatalysts, the highest N_2_O conversion is achieved using samples prepared by electrochemical anodization, especially the F40 sample, showing nanotubular morphology. Indeed, such a TNT layer can convert about 10% of N_2_O after 22 h UVA irradiation. The lowest photocatalytic activities are observed for F0 and SG samples with a conversion about 7.2% and 7.9%, respectively. This is due to two main reasons: (i) these samples exhibit a significantly lower specific surface area compared to nanoporous F20, F40, F60 and F80 and (ii) their thickness is also smaller. Nevertheless, both of these poorly photoactive samples present a higher N_2_O conversion than simple photolysis (6.6%).

### 2.3. Degradation of SMX

The degradation curves of SMX using the samples prepared by electrochemical anodization and the sol–gel method are presented in Figure 4, along with the reusability of the best sample (F20) after 3 repeated runs (Appendix A). Globally, a similar trend is observed between N_2_O reduction and SMX oxidation since the best samples are F20 and F40, i.e., samples with nanotubular morphology. This fact highlights the versatility of anodic TiO_2_ nanotube layers. The degradation extent of SMX after 4 h UVA irradiation reaches 65% and 62% for in the presence of F20 and F40, respectively. The degradation extent decreases to 52% using F60, since the structure is composed of reminiscent nanotubes. The efficiency of the other samples is relatively poor due to the absence of nanotubular morphology. In other words, the number of catalytic sites and the lifetime of charge carriers is probably significantly reduced. The TOC analysis corroborates the degradation curves. Indeed, for F20, F40 and F60, the mineralization extent after 4 h UVA irradiation reaches 11%, 8% and 3%, respectively, while for the other samples, no mineralization takes place. The potential application of anodic TiO_2_ layer in water treatment is performed by the degradation of SMX in secondary effluents of wastewater treatment plants (Appendix A). The degradation efficiency decreases from 65 to 45% due to the negative effects of the wastewater matrix, i.e., the dissolved organic matter that plays the role of ROS scavenger. However, the degradation efficiency is still considered to be satisfactory.

In order to obtain better insights into the degradation mechanism using F20, the SMX degradation by-products are identified using LC-MS. The transformation products of SMX are presented in Appendix A. The initial SMX molecule displayed a peak at [M + H]^+^ = 254.0592 and a sodium adduct [M + Na]^+^ = 276.0410 and [M − H]^−^ = 252.0441. Different degradation pathways are proposed considering the identification of SMX by-products (Figure 5). The formation of P1 occurs through the photo-isomerization of the isoxazole ring, which is also the dominant pathway in the degradation of SMX under UV-mediated degradation [36]. Electrophilic reaction on the aromatic ring leads to the formation of hydroxylated (P2) and dihydroxylated (P3) products, while oxidation of the double bond at the isoxazole ring produced P4. The cleavage of sulfonamide bond by the hydroxyl radical leads to the formation of 3-amino-5-methylisoxazole (P5) and sulfanilic acid (P4). Isoxazole ring rearrangement leads to the formation of P7 that can be further oxidized on the amine ring in the presence of a hydroxyl radical, leading to the formation of P7 and P8. The degradation mechanism confirms the predominant role of hydroxyl radicals.

## 3. Materials and Methods

### 3.1. Preparation and Characterization of Photocatalysts

The TNT layers are prepared using a similar procedure reported in one of our previous study [34]. Briefly, a disk of 4 cm in diameter of titanium foil (Merck, Darmstadt, Germany, 99.7% with 0.127 mm thickness) is used as a working electrode and dropped in fluoride-based electrolyte based on glycerol. The counter electrode is also a 4 cm diameter Ti foil. The distance between the two electrodes is set at 1.5 cm and electrochemical anodization is performed at different applied voltages from 0 to 80 V (with a 20 V step) for 100 min. The current intensity is kept at 5 A during the procedure. After rinsing the as-prepared TNT followed by annealing at 400 °C for 1 h, TNT layers labelled F0, F20, F40, F60 and F80 are obtained. 

For comparison, TiO_2_ sol–gel films are prepared. To this end, titanium isopropoxide (97.0%; Merck, Darmstadt, Germany) is added to isopropanol (reagent grade, Slavus, Bratislava, Slovakia) containing acetic acid (99.0%, Slavus, Bratislava, Slovakia) and Triton^®^ X-100 (Merck, Darmstadt, Germany) as chelating and structure directing agents, respectively, thus a 0.2 M Ti alkoxide sol–gel is obtained. The sol–gel is deposited by spin-coating (Ossila Ltd.) at 2000 rpm on Si wafer (University Wafer Inc., South Boston, MA, USA) with a diameter of 4 cm. Six-layer films are prepared with intermediate annealing at 300 °C for 10 min and final annealing at 450 °C for 1 h. The TiO_2_ sol–gel films are labelled SG. 

The TNT and sol–gel layers are characterized by DRS (Shimadzu UV-2600, IRS-2600 Plus) and XRD (Rigaku SmartLab) to control the optical energy band gap (*E_g_*) and the crystalline phase composition. The optical energy band gap is determined using the Kubelka-Munk theory (Equation (1)) by assuming an indirect band gap, i.e., *z* = 2. The value of *E_g_* is determined by the linear extrapolation of the plot with the *x*-axis. Further characterization details of these reproducible nanomaterials are provided in our previous publications [20,34,35].

(F(*R*) *hν*)^1/*z*^ = *B* (*hν* − *E_g_*)(1)

where F^®^ is the absorption spectra, *h* is the Planck constant, *E_g_* is the energy band gap, *z* is a factor depending on the type of *E_g_*, *ν* is the incident light frequency and *B* is a constant.

### 3.2. Degradation of N_2_O in the Gas Phase

The photocatalytic decomposition of gaseous N_2_O is performed in a custom-made stainless-steel photo-reactor (Figure 6), where the photocatalytic layer is placed at the bottom. After that, the reactor is closed and filled with a N_2_O/He mixture and pressurized to 1.5 bar (pressure is controlled during the whole experiment). The initial N_2_O concentration is set at 1030 ppm. The irradiation is generated by a UVA source (UVP Pen-Ray, 8 W Hg lamp; *λ*_max_ = 365 nm) situated at the top of a photo-reactor, through the quartz glass visor. The N_2_O concentration is measured using a GC/BID (Gas Chromatography coupled with Barrier discharge Ionization detector, Shimadzu Tracera GC 2010 Plus) in two hour-intervals for 22 h. Each experiment is repeated to check the reproducibility. The conversion of N_2_O ($R_{N_{2}O}$) is calculated using Equation (1), where $x_{N_{2}O}^{0}$ is the initial mole fraction of N_2_O and $x_{N_{2}O}$ is the mole fraction at different times during the photocatalytic reaction.
(2)$$R_{N_{2}O}=\frac{x_{N_{2}O}^{0}-x_{N_{2}O}}{x_{N_{2}O}^{0}}$$

### 3.3. Degradation of SMX in Water

Concerning the photocatalytic degradation of SMX solution (50 µM), the photocatalytic layers are placed at the bottom of a homemade photo-reactor equipped of four UVA lamps at the top (Sylvania F15W/350BL; 1.9 mW cm^−2^ in the range 290–400 nm). Prior to turning on the lamps, the initial pH of SMX solution is adjusted at 7 using HClO_4_ and NaOH. The photocatalytic degradation is performed under constant air bubbling for 4 h and 500 µL is sampled out every 30 min (filtration through 0.45 µm PTFE filter and quenching into 100 µL methanol). The concentration of SMX is analyzed by HPLC (Shimadzu Nexera XR LC-20AD) equipped with a C18 column (Agilent, EC 250/4.6 nucleodur 100/5). The mobile phase is a mixture of MeOH/H_2_O in gradient mode from 40:60 (*v*/*v*) to 95:5 (*v*/*v*), which is achieved in 10 min. The detection wavelength of SMX is fixed at 268 nm. It is worth noting that the degradation experiments are repeated three times to control the reusability of the photocatalytic layers and the variations in degradation efficiency do not exceed 5% (Appendix A).

In addition, to highlight the significance of the present study for potential industrial application, the degradation of SMX is performed in the secondary effluent of municipal wastewater treatment plants collected in Clermont-Ferrand, France. Prior to being used as a wastewater matrix, it is filtered through a 0.45 µm PTFE membrane and further analyzed by total organic analysis (Shimadzu, TOC-L) and ionic chromatography (Thermo Scientific, ICS 5000). The data are presented in Table 1.

The identification of SMX transformation by-products is obtained by ultra-high performance liquid chromatography (UHPLC) coupled with high-resolution mass spectrometry (HRMS) performed on an Orbitrap Q-Exactive (Thermo scientific). The column is a Phenomenex Kinetex C18 (1.7 μm × 100 Å; 100 × 2.1 mm) and the temperature is set at 30 °C. The initial gradient is 5% ACN and 95% water acidified with 1% formic acid, followed by a linear gradient to 99% ACN within 8.5 min and kept constant during 1 min. The flow rate is 0.45 mL min^−1^ and the injection volume is 5 μL. Ionization is set to 3.2 kV (ESI+) and 3.0 kV (ESI^−^).

## 4. Conclusions

The preparation of different TiO_2_ nanostructures including nanotube layers and thin films has been performed using electrochemical anodization and the sol–gel method, respectively. Their performance in versatile photocatalytic applications has been assessed by investigating their ability to degrade gaseous and aqueous pollutants under UVA light, i.e., N_2_O and sulfamethoxazole. The best samples in both the photocatalytic reactions are those showing a nanotubular morphology (F20 and F40), probably due to their high specific surface area and excellent charge carriers transport, compared to others samples (F0, F60, F80 and SG) that show non-ordered and less porous nanostructures. In addition, the degradation mechanism of sulfamethoxazole has been proposed and has highlighted the crucial role of hydroxyl radicals. This study is significant since it proves the photocatalytic versatility of anodic TiO_2_ nanotube layers in different applications such as the remediation of air and water. Indeed, the degradation of sulfamethoxazole has been tested in secondary effluents from a wastewater treatment plant and has confirmed the efficiency of the anodic TiO_2_ nanotube layers.

## Figures and Tables

**Figure 1 molecules-27-08959-f001:**
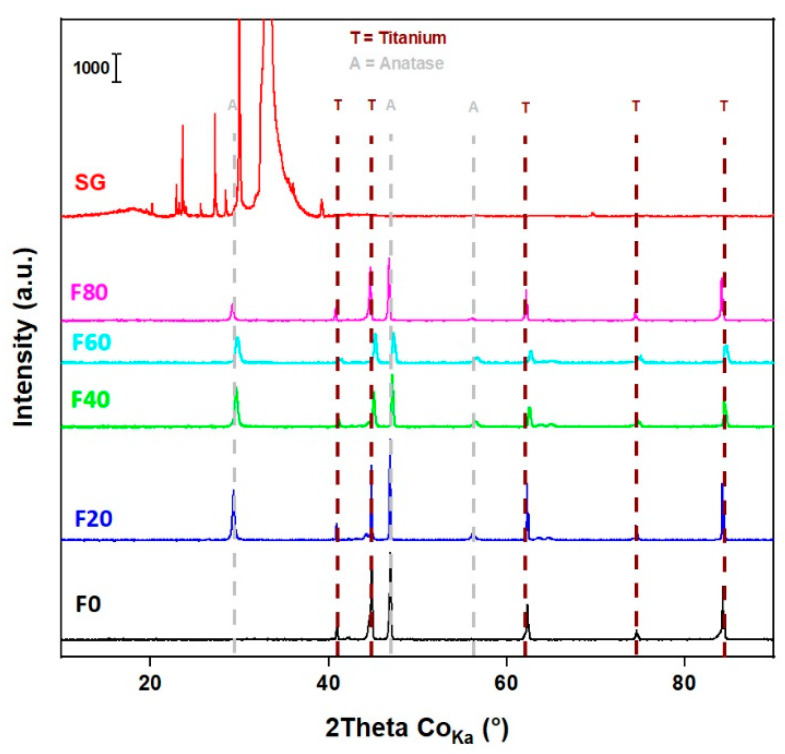
XRD of the TiO_2_ nanostructures.

**Figure 2 molecules-27-08959-f002:**
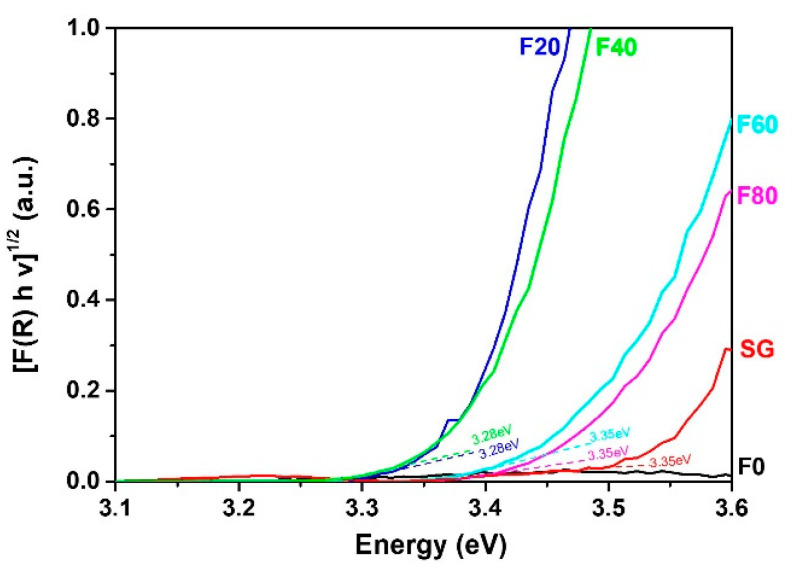
UV-visible DRS of the TiO_2_ nanostructures.

**Figure 3 molecules-27-08959-f003:**
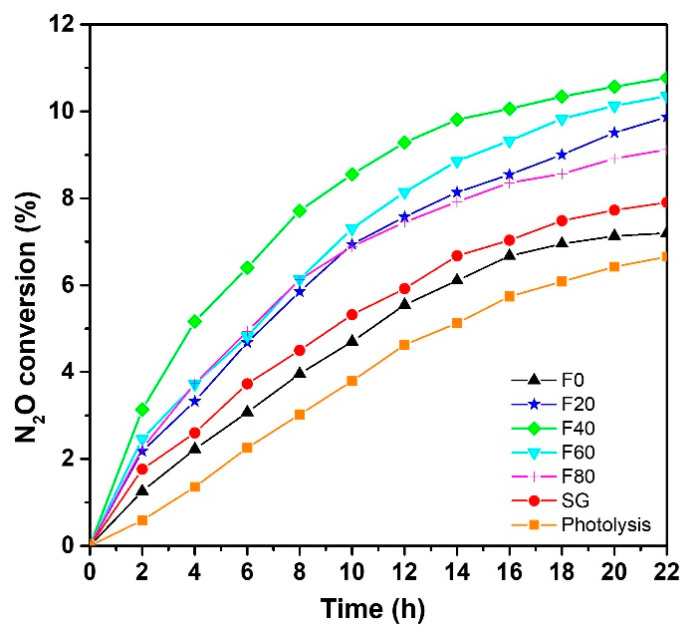
N_2_O gas conversion under UVA light.

**Figure 4 molecules-27-08959-f004:**
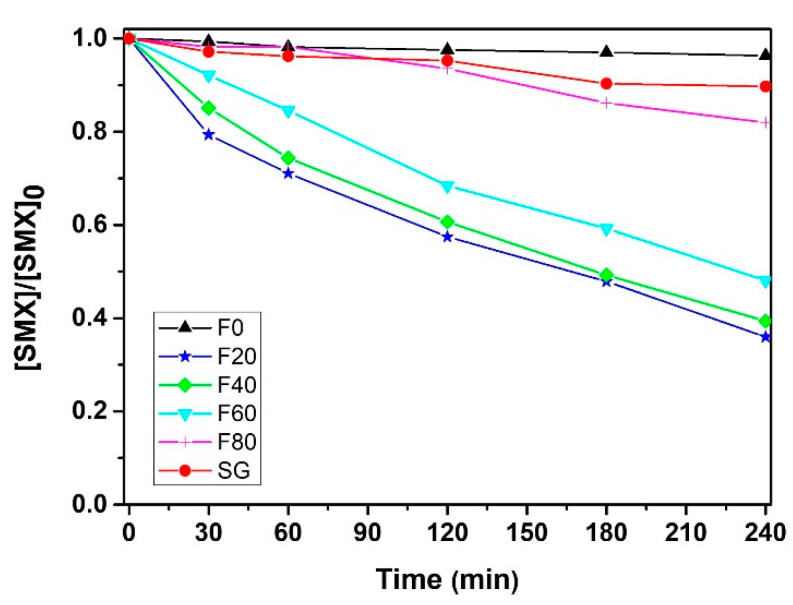
Degradation of SMX under UVA light.

**Figure 5 molecules-27-08959-f005:**
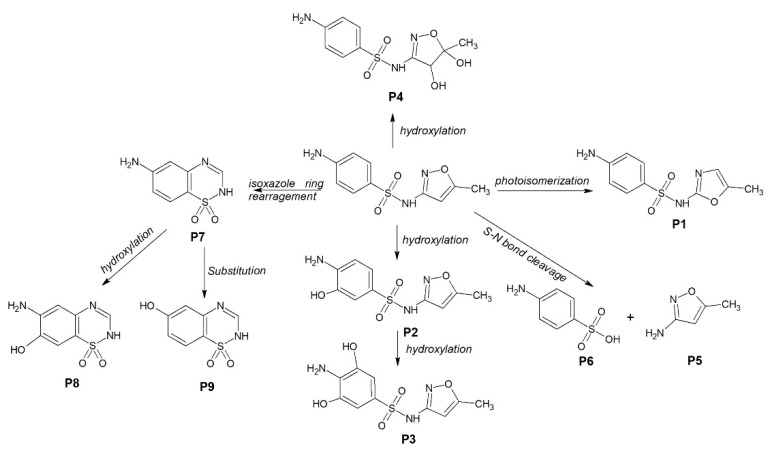
Proposed mechanism of SMX degradation under UVA light in the presence of anodic TiO_2_ nanotube layer.

**Figure 6 molecules-27-08959-f006:**
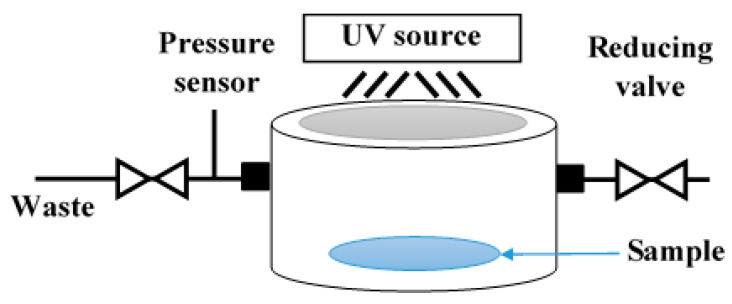
Scheme of the photo-reactor used to decompose N_2_O gas.

**Table 1 molecules-27-08959-t001:** Concentration of inorganic ions and inorganic and organic carbon.

Species	Concentration (mg L^−1^)
Cl^−^	100
NO_3_^−^	59
SO_4_^2−^	53
PO_4_^3−^	<LOD ^1^
Na^+^	131
NH_4_^+^	4
K^+^	40
Mg^2+^	9
Ca^2+^	25
Inorganic C	47
Organic C	7

^1^ Under the Limit of Detection.

## Data Availability

The data are contained within the present manuscript and Appendix A.

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
