# Peer review of "Anodic TiO2 Nanotube Layers for Wastewater and Air Treatments: Assessment of Performance Using Sulfamethoxazole Degradation and N2O Reduction"

_molecules, 2022, doi:10.3390/molecules27248959_

Round 1

Reviewer 1 Report

The Manuscript submitted by Sihor et al, entitled Anodic TiO2 nanotube layers for wastewater and air treatments: 2 Assessment of performance using sulfamethoxazole degrada- 3 tion and N2O reduction, in general is very poor and need more characterization and experimental work.

Here some suggestions that will help the work to be more convincing.

1- The SEM pictures for both TNT and SG are very bad, there is no evidence for the structure you get.

It is mandatory to proof you structure.

2- There are a lot of characterization are missing, like XRD, N2 adsorption, TEM. ( Very important and mandatory)

3- to complete your application on degradation of SMX, there are two experiments are missing. The first one is scavenger test which help you to detect the active species responsible for the oxidation process, while the second one is the recyclability test.

Author Response

Reviewer No. 1

The Manuscript submitted by Sihor et al, entitled Anodic TiO2 nanotube layers for wastewater and air treatments: Assessment of performance using sulfamethoxazole degradation and N2O reduction, in general is very poor and need more characterization and experimental work.

Here some suggestions that will help the work to be more convincing.

 1- The SEM pictures for both TNT and SG are very bad, there is no evidence for the structure you get.

It is mandatory to proof you structure.

Answer: Indeed, the SEM pictures are not well resolved. To proof the structure of our materials, we decided to delete this picture and to report some of our published works in the main text, since the same experimental procedure was used to prepare the different TNT and SG layers. It was already mentioned in the main text. Please, refer to these works.

Ref. 20 - Hanif, M.B.; Sihor, M.; Liapun, V.; Makarov, H.; Monfort, O.; Motola, M. Porous vs. Nanotubular Anodic TiO2: Does the Morphology Really Matters for the Photodegradation of Caffeine? Coatings 2022, 12, 1–12, doi:10.3390/coatings12071002.

Ref. 34 - Sihor, M.; Hanif, M.B.; Thirunavukkarasu, G.K.; Liapun, V.; Edelmannova, M.F.; Roch, T.; Satrapinskyy, L.; Pleceník, T.; Rauf, S.; Hensel, K.; et al. Anodization of Large Area Ti: A Versatile Material for Caffeine Photodegradation and Hydrogen Production. Catal. Sci. Technol. 2022, 12, 5045–5052, doi:10.1039/d2cy00593j.

Ref. 35 - Monfort, O.; Roch, T.; Gregor, M.; Satrapinskyy, L.; Raptis, D.; Lianos, P.; Plesch, G. Photooxidative Properties of Various BiVO4/TiO2 Layered Composite Films and Study of Their Photocatalytic Mechanism in Pollutant Degradation. J. Environ. Chem. Eng. 2017, 5, 5143–5149, doi:10.1016/j.jece.2017.09.050.

2- There are a lot of characterization are missing, like XRD, N2 adsorption, TEM. (Very important and mandatory)

Answer: The XRD has been already performed (and based on Reviewer No.2 comment, we have moved the Figure from supplementary materials file to the main text). For further characterization, as previously explained in the previous comment, we invite you to read our previous works.

We have to stress that the focus of this work is to highlight for the first time the use of TNT in the degradation of N2O gas along with the degradation of SMX, thus showing the versatility in photocatalytic application. We believed we have reached this goal.

3- to complete your application on degradation of SMX, there are two experiments are missing. The first one is scavenger test which help you to detect the active species responsible for the oxidation process, while the second one is the recyclability test.

Answer: It is well known that the main active species in TiO2 photocatalysis are hydroxyl radicals (see introduction). These experiments have been already performed in the literature as well as in our previous work. We think that to proceed with such experiments will not bring novelty but overload the current manuscript. However, the LC-MS has shown the degradation byproducts, thus highlighting the mechanism of degradation including the attack of hydroxyl radicals.

In addition, the recyclability tests have been performed. See section 2.2: “each experiment is repeated to check the reproducibility” (in section 2.3, we forget to mention it, but now the text has been corrected). The results are that we do not observe significant decrease in the photocatalytic efficiency after repeated runs (variations are not exceeding 5%). Figure S1 shows the recyclability of our materials.

Reviewer 2 Report

Anodic TiO2 nanotube layers for wastewater and air treatments: Assessment of performance using sulfamethoxazole degradation and N2O reduction 

The article is well written and easy to read. The author provides ample background information and explains the methodology used in a clear and concise manner. The results of the study are presented in an organized and logical form. The article concludes with a thoughtful discussion of the implications of the findings. Here are my comments.

1.     Please explain more in the introduction section.

2.     The XRD pattern can be changed to the main manuscript instead of in the supplementary information.

3.     Figure 3 have to be modified with the straight line interpolation to show the bandgap values exactly.

4.     Please explain briefly how the bandgap was calculated, also add the discussion in the manuscript.

Author Response

Reviewer No.2

Anodic TiO2 nanotube layers for wastewater and air treatments: Assessment of performance using sulfamethoxazole degradation and N2O reduction

 The article is well written and easy to read. The author provides ample background information and explains the methodology used in a clear and concise manner. The results of the study are presented in an organized and logical form. The article concludes with a thoughtful discussion of the implications of the findings. Here are my comments.

  1. Please explain more in the introduction section.

Answer: the introduction has been slightly more elaborated.

  1. The XRD pattern can be changed to the main manuscript instead of in the supplementary information.

Answer: the XRD results has been inserted in the main text.

  1. Figure 3 have to be modified with the straight line interpolation to show the bandgap values exactly.

Answer: the Figure 3 has been modified, so the bandgap values can be read.

  1. Please explain briefly how the bandgap was calculated, also add the discussion in the manuscript.

Answer: the discussion about the bandgap calculation has been added in the main text (see section 2.1).

Round 2

Reviewer 1 Report

Thanks to the authors for the modification they did.

please, cite the following paper:

1- Silver decorated TiO2/g-C3N4 bifunctional nanocomposites for photocatalytic elimination of water pollutants under UV and artificial solar light. doi.org/10.1016/j.rineng.2022.100470

2- Magnetic TiO2/CoFe2O4 Photocatalysts for Degradation of Organic Dyes and Pharmaceuticals without Oxidants. /doi.org/10.3390/nano12193290